# Peer review of "Developing an Enzyme-Assisted Derivatization Method for Analysis of C27 Bile Alcohols and Acids by Electrospray Ionization-Mass Spectrometry"

_molecules, 2019, doi:10.3390/molecules24030597_

Round 1

Reviewer 1 Report

The topic is very relevant, considering the importance of sterolomics (and the growing realization of this), and the many challenges related to the analysis of this diverse group of molecules. 

This communication introduces a new approach allowing for an expansion of molecules "eligible" for charge-tagging, which has the potential for being a useful tool for sterol-analysts.

The authors have written a very to-the-point report on these initial findings, along with a condensed but informative introduction.

The authors are frank about the preliminary nature and current stage limitations but is nonetheless a highly relevant paper.

I would, however, like to see how the detection limits compare to other related methods/approaches/analytes, and also some lines on the repeatability of the procedure, and prospects moving towards quantification.  

Author Response

Response to Reviewer 1 Comments

Point 1: I would, however, like to see how the detection limits compare to other related methods/approaches/analytes,

Response 1: On page 6, lines 257 -265 we have compared our detection limits to some of those found in the literature, i.e. Johnson et al. 2001 and DeBarber at al. 2014.

Point 2: and also some lines on the repeatability of the procedure, and prospects moving towards quantification

Response 2: As yet we have not performed thorough repeatability studies, this data will be more relevant when isotope-labelled standards become available. Although, not currently commercially available, synthetic methods have been described for the production of isotope-labelled cholestanoic acids and bile alcohols. This is detailed on page 6, lines 264 – 269. 

Reviewer 2 Report

In this communication the authors describe a new study for the enzyme-assisted  derivatization for C27 bile alcohols and acids with a 3α-hydroxy-5β-hydrogen stereochemistry. The authors indicated that their method still requires further optimization, particularly with respect of the GP-derivatisation step which only gave a 45 – 60% yield.

The authors found their method to be considerable sensitivity producing GP-derivatives in moderate yields. The authors found that by performing ESI-MS and CID-MS3 analyses they obtained better series of product ions providing a significant advantage over conventional MS/MS analysis of unconjugated acids, in which few product ions are observed.

The authors also applied their sterol profiling method to bile acids and alcohols derivatized with [2H0]GP  after 3α-HSD treatment and oxysterols, cholestenoic and cholenoic acids derivatized with [2H5]GP  after cholesterol oxidase treatment, or vice versa, into a single LC-MS run.

This communication is well written. However, the MS/MS nomenclature has been mangled and not respected. It should be rehashed and rewritten with the proper MS terminology.

1. In your Abstract, lines 20-23, you have written:

However, as found by others, derivatization of the resultant 3-oxo group with a hydrazine reagent does not go to completion in the absence of a conjugating double bond in the sterol structure. Never-the-less, Girard P derivatives of bile alcohols and C27 acids give an intense [M]+ ion upon electrospray ionization and informative fragmentation spectra. The method shows promise for analysis of bile alcohols and 3α-hydroxy-5β-C27-acids, enhancing the range of sterols that can be analyzed at high sensitivity in sterolomic studies.

QUERY: [M]+, what do you mean exactly?

2. In page 2 of your pdf, lines 61-82.

Correct Une et al for Une et al.

3. In page 4 of your pdf, lines 135. Can you explain how do you get the [M-Py]+ ions?→

QUERY: Do not be shy, explain exactly the benefits of introducing this Girard derivative.

QUERY: I think it is important that you pinpoint the formation of the introduction of the Girad derivatized pyridine group that in fact helps to cationize the original molecule, by virtue of the + charge gained it also facilitate the ESI process.

4. General comment:

In you figures you represent your MS3 as [M]+→[M-79]+→[M-79-H2O]+ Rationally, using IUPAC nomenclature you are expected to put the correct m/z values underneath each notations !

THAT SHOULD ALSO BE INCLUDED THROUGHOUT YOUR TEXT.

5. Where is your List of Legends?

6. In you figures 1- 5, why are you showing the RIC?  It is essential that you also show the ESI-MS showing their [M]+ derivatives full scans.

Author Response

Response to Reviewer 2 Comments

Point 1:. QUERY: [M]+, what do you mean exactly?

Response 1: [M]+ refers to the molecular ion. This is now stated on lines 22 – 23 on page 1.

Point 2: Correct Une et al for Une et al.

Response 2: Throughout we have now adopted the convention et al.

Point 3: In page 4 of your pdf, lines 135. Can you explain how do you get the [M-Py]+ ions?

Response 3: We have added the prompt “see Scheme 1” to line 135. Scheme 1 explains the formation of [M-Py]+ ions.

QUERY: I think it is important that you pinpoint the formation of the introduction of the Girad derivatized pyridine group that in fact helps to cationize the original molecule, by virtue of the + charge gained it also facilitate the ESI process.

Response: On page 2, lines 59 – 61, we write “the 3-oxo group is derivatised with the positively charged Girard hydrazine reagent, introducing a charge-tag to the target analyte and improving sensitivity in ESI-MS”.

Point 4: General comment:

In you figures you represent your MS3 as [M]+→[M-79]+→[M-79-H2O]+ Rationally, using IUPAC nomenclature you are expected to put the correct m/z values underneath each notations.

Response: We now clarify the m/z values for each of the above ions by reference in every figure caption to Table 1 where the ion-nomenclature is correlated to m/z.

Point 5: Where is your List of Legends?

Response: Legends are embedded within the text.

Point 6: In you figures 1- 5, why are you showing the RIC?  It is essential that you also show the ESI-MS showing their [M]+ derivatives full scans.

Response: The reason to show RICs is to indicated chromatographic separations where possible and the chromatographic peak shape with the combination of column, mobile phase and flow-rate used. We now display mass spectra showing [M]+ ions in Supplemental Figure S2.